# Immunoglobulin G Subclass-Specific Glycosylation Changes in Rheumatoid Arthritis

**DOI:** 10.3390/ijms26199626

**Published:** 2025-10-02

**Authors:** Dániel Szabó, Balázs Gyebrovszki, Eszter Szarka, Felícia Auer, Bernadette Rojkovich, György Nagy, András Telekes, Károly Vékey, László Drahos, András Ács, Gabriella Sármay

**Affiliations:** 1MS Proteomics Research Group, HUN-REN Research Centre for Natural Sciences, 1117 Budapest, Hungary; szabo.daniel@ttk.hu (D.S.); vekey.karoly@gmail.com (K.V.); drahos.laszlo@ttk.hu (L.D.); 2Hevesy György PhD School of Chemistry, Faculty of Science, Eötvös Loránd University, 1117 Budapest, Hungary; 3Department of Immunology, Eötvös Loránd University, 1117 Budapest, Hungary; gyebrovszki.balazs@gmail.com (B.G.); estrogens@gmail.com (E.S.); auer.felicia@mk.uni-pannon.hu (F.A.); 4Translational Glycomics Research Group, University of Pannonia, 8200 Veszprém, Hungary; 5Rheumatology-Rehabilitation Department, Buda Hospital of the Hospitaller Order of Saint John of God, 1027 Budapest, Hungary; rojkovich.b@gmail.com; 6Department of Rheumatology and Immunology, Semmelweis University, 1085 Budapest, Hungary; nagy.gyorgy2@semmelweis.hu; 7Heart and Vascular Centre, Semmelweis University, 1085 Budapest, Hungary; 8Department of Genetics, Cell- and Immunobiology, Semmelweis University, 1085 Budapest, Hungary; 9Department of Oncology, St Lazarus County Hospital, 3100 Salgótarján, Hungary; prof.andras.telekes@gmail.com

**Keywords:** anti-citrullinated protein antibodies, glycosylation, immunoglobulin G isoforms, arthritis, rheumatoid

## Abstract

Rheumatoid arthritis (RA) is the most common inflammatory polyarthritis. In addition, 60–80% of patients express anti-citrullinated protein antibodies (ACPAs), which serve as a diagnostic marker for RA. The effector functions of these autoantibodies can be heavily affected by the N-glycosylation of their Fc region. Here we present a comparison of the Fc N-glycosylation of ACPA IgG to that of non-ACPA IgG from the same patients, and of healthy controls, in an IgG isoform-specific manner. We isolated ACPA and normal serum IgG, digested by trypsin, and separated the resulting peptide mixture by a reversed-phase nanoLC coupled to a Bruker Maxis II Q-TOF, and determined the relative abundance of glycoforms. The paired analysis of galactosylation and sialylation of the IgG subclasses of ACPA and non-ACPA IgG has shown a significant, moderate negative correlation with the inflammatory markers, the level of C-reactive protein (CRP) and erythrocyte sedimentation rate (ESR), as well as with rheuma-factor (RF), but not with the disease activity score (DAS) or cyclic citrullinated peptide specific antibodies (anti-CCP). However, we detected a significant negative correlation between glycosylation and DAS in the non-ACPA IgG fractions. Furthermore, the isoform-specific analysis revealed additional insight into the changes of the glycosylation features of IgG in RA: changes in the frequencies of the bisecting GlcNAc unit between sample groups could be explained by only the IgG1 isoform; while invariance in fucosylation is the result of the superposition of two isoforms with opposite changes. These results highlight the importance of analyzing immunoglobulin glycosylation in an isoform-specific manner.

## 1. Introduction

Rheumatoid arthritis (RA) affects around 0.6–0.8% of the Western world population, predominantly women, with a ratio of approximately 3:1 [1,2,3]. The molecular background of disease development and progression has been extensively studied, and although the pathomechanism is not yet fully elucidated, it has been demonstrated that autoantibodies play a crucial role in the course of the disease [4,5,6,7].

In RA, two main types of autoantibodies are best characterized: RF (rheumatoid factor) and ACPA (anti-citrullinated protein antibodies). The presence of these autoantibodies can predict the clinical development of RA years in advance [8]. RF and ACPA have similar diagnostic sensitivities; however, ACPA are rather specific diagnostic markers and prognostic indicators [9].

Each isoform of IgG ACPA (IgG1, IgG2, IgG3, and IgG4) has a highly conserved N-glycosylation site at position Asn297 within the constant CH2 domain of each of the two heavy chains. IgG3 also contains an additional N-glycosylation site on its SH3 domain [10]. Around 20% of IgG molecules also carry an additional N-glycosylation site in the variable region (Fab glycans), but it is only the Fc glycosylation that influences the antibodies’ effector functions [11]. The glycan structures attached to Asn297 on the Fc region of IgG are highly diverse [12,13] and significantly affect its conformation. This alters its binding to Fcγ receptors and its interactions with other molecules, such as C1q, which both mediate effector functions. Additionally, glycans may stabilize the CH2 domain conformation [14,15,16].

IgG N-glycans are composed of a common core structure containing two N-acetylglucosamine (GlcNAc) and three mannose residues, which can be further extended with fucose, galactose, sialic acid, or a bisecting GlcNAc unit [17,18]. It is not completely clear yet how the various glycoforms modulate the effector function of IgG, although several aspects have been extensively examined [19,20,21,22]. The most prominent change was observed regarding galactosylation. Historically, a decrease in IgG galactosylation was first observed in RA patients [23]. Subsequently, this trend has been observed in other autoimmune diseases, such as myasthenia gravis and systemic lupus erythematosus, and it was confirmed that decreased IgG galactosylation is a hallmark of inflammation. During a flare of disease, the total IgG galactosylation is low, reducing the binding to FcγRIII, thus lowering the threshold for FcγR-activation by allowing pathogenic autoantibodies to cause immune activation [24,25,26,27].

Increased sialylation has been found to extend the serum half-life of monoclonal antibodies [28]. Additionally, it was revealed that the sialylated IgG binds to type 2 Fc receptors, including C-type lectin receptors on dendritic cells and macrophages, as well as to sialic acid-binding immunoglobulin-like lectins [29,30,31,32]. Binding to these type 2 Fc receptors enables sialylated IgG to exert an anti-inflammatory effect by triggering various suppressive pathways.

Regarding fucosylation, it has been demonstrated that the absence of fucose increases FcγRIIIA (Fcγ receptor IIIA) IgG binding, thus enhancing ADCC (antibody-dependent cellular cytotoxicity) and phagocytosis [33].

Less is known about the effect of bisecting GlcNAc, since its presence hinders the connection of a core-fucose; therefore, it is hard to discriminate its effect from the lack of fucosylation [34,35,36].

Glycosylation of antigen-specific IgG was shown to change upon vaccination in some cases from that of the overall IgG (also called bulk IgG) glycosylation profile [37,38]. ACPA glycosylation profile has been compared to the total IgG glycan features in RA [39,40]. These works compared the glycosylation of ACPA IgG1 with the total IgG1 glycoforms. The authors concluded that ACPA display significant changes in Fc galactosylation and fucosylation before the onset of the disease. These modifications towards a more pro-inflammatory phenotype could be involved in driving the disease process. Differences were reported in galactosylation, fucosylation, and sialylation. Differences in glycosylation were also demonstrated between various IgG isoforms [41]. One group presented a comparison of IgG isoform-specific glycosylation of ACPA with that of non-ACPA IgG in the case of a limited number of patients (6 female, 8 male) [42]. Irrespective of disease state, IgG glycosylation is known to be influenced by various factors, such as age and gender [43,44]. A large-scale analysis of 2298 individuals has revealed that age was associated with a significant decrease in galactosylation and an increase in the presence of bisecting GlcNAc [45].

Protein glycosylation influences many biological processes but is difficult to analyze due to the structural heterogeneity of glycan side chains. ACPAs play a major role in the pathomechanism of RA; however, their glycosylation pattern has only been analyzed in a limited number of papers [39,40,42,46,47].

Although, subclass-specific separation of IgG isolated from RA patients has been performed on several occasions in various clinical settings, it has rarely been performed in the case of ACPA due to the complexity of sample preparation [21]. In this paper, we present a comparison of the subclass-specific glycosylation patterns of three different groups: ACPA and non-ACPA IgG fractions separated from the sera of the same RA patients, and IgG obtained from the sera of healthy individuals.

## 2. Results

### 2.1. Overall Trends of Glycosylation of IgG from RA Patients and Healthy Controls

Three sample groups were studied: the citrullinated peptide-specific IgG fraction of RA patients (ACPA), the ACPA-depleted IgG fraction of RA patients (non-ACPA), and IgG of healthy controls (Healthy Control or HC). First, we investigated overall trends of IgG glycosylation in all three sample groups in an extended set of samples ($n_{ACPA}=22,\;n_{non-ACPA}=17,\;n_{HC}=30$) spanning a wide age range (from 27 to 81 years). Subsequently, we have refined the comparison between the sample groups in an age-matched subset of the samples ($n_{ACPA}=16,\;n_{non-ACPA}=12,\;n_{HC}=10$); where the average age is approximately 58 years in each group. Details of the participants as well as further clinical details of the RA patients are summarized in Table A1.

We included only those glycoforms that were reliably identified (detected in at least two-thirds of the samples); thus, nineteen glycoforms were investigated in the case of both IgG1 and IgG2, while eight N-glycopeptides of IgG3/4 were studied, due to their generally lower abundance. Of the nineteen studied glycans, four were completely non-galactosylated, and seven lacked the core-fucose moiety. Seven glycans were sialylated, and seven contained a bisecting GlcNAc unit. The investigated glycan structures, along with their average relative intensities in the age-matched ACPA samples, are illustrated in an IgG subclass-specific manner in Figure 1.

A negative correlation between the level of galactosylation and age is observed; across all isoforms and all sample groups (i.e., both healthy and RA, Figure 2a). Our results show that the correlation is somewhat weaker in the case of healthy controls than in RA patients (${R^{2}}_{ACPA,Gal}=0.52$, ${R^{2}}_{non-ACPA,Gal}=0.49$, ${R^{2}}_{HC,Gal}=0.20$). Similar relationships hold true for all three isoforms individually as well (Figure 2b–d); however, the IgG3/4 isoform displays weaker correlations irrespective of the sample group.

A similar trend could be observed in the case of the level of sialylation (Figure A1), which decreases with the age of the individuals as well. However, the correlations between sialylation and age are lower than in the case of the level of galactosylation (${R^{2}}_{ACPA,Sia}=0.39$, ${R^{2}}_{non-ACPA,Sia}=0.39$, ${R^{2}}_{HC,Sia}=0.16$, across all IgG isoforms). This relationship is more pronounced for IgG1, while the correlation is weaker for the other isoforms. It should be noted that the degree of galactosylation and the degree of sialylation are highly correlated in all sample groups (Figure 3); thus, the similarities in their trends are not surprising.

In contrast, no meaningful linear correlation was observed between the age of the individuals with the level of fucosylation, nor with the frequency of the occurrence of bisecting GlcNAc. Furthermore, there was no solid evidence for the core fucosylation-hindering effect of the presence of the bisecting GlcNAc unit: only a weak inverse correlation could be observed between the fucosylation level and the frequency of the bisecting unit ($R^{2}\leq 0.19$).

### 2.2. Comparative Results Between Age-Matched Groups

Due to the age-dependent nature of some of the glycosylation features, an age-matched set of samples was selected to facilitate the comparative analysis of IgG glycosylation between the sample groups. The mean age of the selected healthy controls was 58.3 years ($n_{HC}=10$); for the RA patients with ACPA IgG samples, it was 58.6 years ($n_{ACPA}=17$). For those RA patients who had non-ACPA IgG samples available, the average age was 57.9 years ($n_{non-ACPA}=12$). The results presented in the following section were all obtained using this age-matched set of samples. For the RA patients, who had non-ACPA IgG samples available, a paired comparison was made as well between their ACPA IgG and non-ACPA IgG results; such comparisons will be referred to as those of ‘paired samples’.

The degree of galactosylation is one of the most studied aspects of IgG N-glycosylation in the literature, with the consensus being that RA patients display lower galactosylation compared to healthy individuals. We suggest that this is partly due to age differences (Figure 2). Within our experiments, we found a small but significant decrease in the degree of galactosylation of the ACPA IgG compared to that of healthy controls. This decrease is most evident for IgG1 (p=0.052) and IgG3/4 (p=0.046) and is completely absent in IgG2 (Figure 4). Comparison between the paired samples reveals that ACPA IgG displays decreased galactosylation compared to the non-ACPA fraction for all isoforms, except for IgG2.

The degree of sialylation, despite its relatively strong correlation with the degree of galactosylation, did not display similar trends when the RA and healthy samples were compared (Figure 5). The largest difference observed in this regard was an increase in the degree of sialylation in the non-ACPA fraction as compared to healthy IgG, which is not significant. However, a considerably lower degree of sialylation was detected in the ACPA IgG as compared to the non-ACPA IgG fractions, which was consistent across isoforms. (Figure 5b–d). This decreased sialylation is reinforced by the paired sample comparison of the RA patients, showing a significant change for all isoforms.

Although the degree of fucosylation showed very little variability between the sample groups for the overall IgG content, an interesting trend could be observed on the isoform-specific level. The degree of fucosylation of the IgG1 isoform is significantly higher in the ACPA sample group than in the healthy controls (p=0.050). In contrast, fucosylation of the IgG2 isoform is significantly lower in the ACPA sample group than in healthy controls (p=0.009) (Figure 6). The comparison between the paired samples reveals a very similar trend between isoforms: while the degree of fucosylation on the level of all isoforms decreases in the non-ACPA fraction compared to the ACPA (Figure 6a) driven by the IgG1 isoform (Figure 6b), it increases for the IgG2 isoform for all but one patient (Figure 6c). Note that, as all observed IgG3/4 glycoforms were fucosylated, this comparison is irrelevant for these isoforms.

The non-ACPA fraction showed a significant difference in the frequency of the bisecting GlcNAc unit, compared to both the ACPA (p=0.015) and the healthy control (p=0.006) sample groups but only in the case of the IgG1 isoform. Due to the high abundance of this isoform in serum, this effect is detectable in the overall IgG glycosylation as well. Interestingly in the case of IgG2, only the non-ACPA and healthy control group appear slightly different, while the IgG3/4 isoform shows no notable differences between the sample groups (Figure 7). However, on the level of the paired samples, an increase can also be detected for the remaining two isoforms (Figure 7c,d), although to a much smaller degree, compared to the effect observed for IgG1.

### 2.3. Correlations Between the IgG Fc Glycoforms and the Clinical Parameters of RA Patients

Isoform-specific glycosylation and the relative abundance of individual glycans in ACPA and non-ACPA (flow-through fraction) IgG from RA patients were correlated with clinical parameters. The range of clinical parameters was measured at the time of blood sample collection. Galactosylation and sialylation of IgG ACPA, as well as IgG1 and IgG2 isoforms, showed a moderate, significant negative correlation with age, C-reactive protein (CRP), RF, and erythrocyte sedimentation rate (ESR) values. In contrast, no correlation was observed with DAS and anti-CCP titers in the serum. IgG3/4 galactosylation and sialylation as well as sialylation of IgG2, were negatively correlated only with age and CRP. Conversely, we did not find any correlation of the clinical parameters with the presence of terminal fucose or the bisecting GlcNAc unit. Glycosylation of ACPA and non-ACPA IgG from the same patients displayed a similar pattern of negative correlations, except that the latter (galactosylation of IgG1 and IgG2, and sialylation of IgG3/4) also negatively correlated with DAS (Figure 8a).

We have performed the same analysis with the relative abundance of individual glycans and identified some dominant glycans that followed a similar pattern of correlation with the clinical parameters, for example, IgG1–N4H5S1 with age, DAS, and ESR, IgG1–N5H5S1F1 with age, CRP, and ESR (Figure 8b). Interestingly, we observed a significant moderate positive correlation between the relative abundances of N4H3F1, N4H4F1, N4H4S1F1, and N5H3F1 glycans in all of IgG3/4 ACPA and non-ACPA samples with DAS only (Figure 8b).

## 3. Discussion

IgG galactosylation is the most frequently discussed glycosylation feature [27,44,48,49], both in general and in relation to RA, and it is suggested to decrease in inflammatory diseases. Large-scale studies have also well documented its decrease with age, regardless of inflammation [44,45]. When the age of the individuals is not restricted, our results are consistent with this overall trend; however, the examination of age-matched samples gives further insight.

Interestingly, on the isoform level, IgG2 expresses practically no changes in galactosylation, comparing the ACPA and the healthy control sample group. This is also consistent with the results of Plomp et al., who investigated a cohort of 1826 individuals, and the subclass-specific analysis revealed that galactosylation (and sialylation) of IgG2 shows a weaker association with metabolic markers and inflammation compared to the other isotypes [41]. Wieczorek et al. presented that the plasma IgG2 glycosylation profile is distinctly different from IgG1 and IgG3 in epithelial ovarian cancer (EOC) patients [50]. IgG2 was previously reported to exhibit the lowest affinity for FcγRs, and it is a weak inducer of ADCC compared to other IgG subclasses [34]. As IgG in the bloodstream consists of only 20–30% IgG2, the dominant isoform IgG1 (60–70%) can easily cover up these differences during total IgG analysis [51]. In our previous paper, we examined IgG samples to assess their capability of inducing TNFα production. Our results demonstrated a lower level of galactosylation and sialylation of ACPA samples compared to non-ACPA and healthy IgG, showing a correlation with the TNFα-inducing capacity, in a sample group where the age of the patients differed on a larger scale [46].

The level of sialylation also displays a decreasing trend with age of the individuals, but the correlation is lower than in the case of galactosylation. In the age-matched groups, no real difference occurred between the ACPA and the healthy group; however, the non-ACPA fraction showed a consistently higher level of sialylation than the ACPA samples, in the case of all isoforms. The higher sialylation of non-ACPA IgG in our cohort compared to the healthy group might be explained by the latter’s heterogeneity; some individuals may have experienced unidentified inflammation. General opinion suggests a decrease in sialylation in inflammatory diseases (including RA) [52,53,54]. The consistently lower level of sialylation of ACPA compared to non-ACPA IgG from the same patients in our cohort is in line with previous findings suggesting that decreased sialylation of ACPA could represent a functional switch, converting IgG molecules from an anti-inflammatory to a pro-inflammatory state in autoantibody-producing cells [21].

To find out if altered glycosylation of ACPA is associated with disease activity, the relative abundance of glycosylation in paired samples of ACPA and non-ACPA IgG was analyzed for correlations with age and clinical parameters of the patients. We observed a significant moderate inverse correlation between galactosylation and sialylation of all tested IgG1 and IgG2 samples and the markers of inflammation, CRP, ESR, and RF, as well as age. IgG3/4 glycosylation only correlated with age and CRP. CRP is a marker and regulator of systemic inflammation in RA and also plays a role in disease progression. Although its values may fluctuate, it is a good indicator of disease severity [55]. ESR is less reliable as a sole indicator; it may be normal when disease activity is high. RF levels as well as anti-CCP levels in RA can vary over time and may not always correlate with disease activity or severity [56,57].

We could not detect a significant correlation between glycosylation and anti-CCP levels in any of the samples; thus, glycosylation had no impact on autoantibody production. Glycosylation of ACPA and non-ACPA IgG differed regarding the correlation with DAS. The galactosylation and sialylation of non-ACPA IgG in the flow-through fractions, but not of ACPA IgG, showed a significant negative correlation with DAS. Blöchl et al. recently published a similar conclusion: IgG1 galactosylation and sialylation were found to be negatively correlated with ESR and DAS in total IgG, but not in ACPA IgG. A possible explanation might be the presence of autoantibodies with different, unknown specificities in the non-ACPA IgG fractions [47].

By analyzing the correlation between the relative intensity of individual glycans and clinical parameters, we identified dominant glycans that exhibited a similar correlation pattern to the IgG1 isoform. These included correlation of IgG1N4H5S1 with age, DAS, and ESR; and IgG1–N5H5S1F1 with age, CRP, and ESR.

The role of fucosylation in the pathomechanism of RA is still a controversial topic. Gornik et al. found a 40% increase in the level of fucosylation in an earlier study comparing bulk IgG of RA patients with healthy controls [58]. Later, Zhipeng et al. compared the N-glycans of IgG obtained from the sera of 44 RA patients and 30 healthy controls and presented a similar trend [59]. In the case of ACPA, Scherer et al. detected no significant overall difference between ACPA IgG1 and total serum IgG1, while the groups of Rombouts and Lundström described a slight increase in the level of fucosylated glycoforms on ACPA IgG1 [39,40,42]. Our results suggest no significant correlation between the level of fucosylation and the age of the individuals. Even though, the overall fucosylation did not show marked differences between the studied sample groups, comparing fucosylation on the isoform level revealed interesting differences: ACPA IgG1 shows a significantly higher degree of fucosylation than the IgG1 isolated from healthy controls; while, in contrast, ACPA IgG2 fucosylation changes in the opposite direction and shows a lower level of fucosylation compared to the IgG2 derived from the healthy group. This interesting observation further elaborates the difference between IgG subclasses, which can initiate a response on the functional level. Studies have suggested that the fucosylation of IgG affects receptor interactions with Fcγ receptor IIIa (FcγRIIIa). Mizushima et al. and Coillie et al. have confirmed that the lack of fucose residue on the N-glycans of the Fc portion of IgG results in improved affinity for FcγRIIIa, which leads to dramatic enhancement of antibody-dependent cellular cytotoxicity (ADCC) [60,61]. This presents in an even more compelling context for the fact that in the case of IgG3/4, we detected only fucosylated glycoforms. Additionally, the relative intensity of IgG3/4 glycans: N4H3F1, N4H4F1, N4H4S1F1, and N5H3F1 showed a positive correlation with DAS in both ACPA and non-ACPA IgG fractions. While the initial key points of subclass-specific fucosylation are still unclear, the presented differences further emphasize the need for independent analysis of IgG subclasses.

Little is known about the importance of the biological implications of glycan moieties containing bisecting GlcNAc [34]. The presence of bisecting GlcNAc in Fc glycans has been described to induce antibody-dependent cell-mediated cytotoxicity [62,63]. Furthermore, Nakano et al. reported that the bisecting GlcNAc suppresses the attachment of terminal epitopes, such as sialic acid or fucose [64]. Our results show no correlation between the level of bisecting GlcNAc and the degree of sialylation, while no conclusion can be drawn regarding the fucosylation hindering effect of the bisecting GlcNAc unit, since our study did not include any antenna fucosylated glycoforms. The level of bisecting GlcNAc showed no linear relationship with the age of the individuals, contrasting the work of Pučić et al., which suggests a significant increase in bisecting GlcNAc is associated with age [45]. Interestingly, in the age-matched groups we have found that the highest level of bisecting GlcNAc occurs in the non-ACPA fraction. This was presented most prominently in the IgG1 subclass. Lundström et al. described a lower level of bisecting GlcNAc in ACPA IgG1 compared to the non-ACPA IgG1, which is consistent with our results [42]. The comparison of results of different articles found in the literature is limited by various factors. First, large sample studies show that glycosylation features change with age; however, age-matched groups are not consistently applied, which can greatly influence the results, as it was previously mentioned. Second, sample preparation methods can lead to loss or underrepresentation of certain glycoforms. One such example is sialic acid, which is well-known for its labile nature [65]. And third, IgG isoforms have several genetic variants, and the subtle differences in the amino acid sequence can result in glycopeptides with identical mass, which cannot be discriminated by mass spectrometry. Articles may also differ in considering sequences to a certain isoform.

## 4. Materials and Methods

### 4.1. Collection of Blood Samples

Blood samples were collected from RA patients diagnosed according to the revised classification criteria of the American College of Rheumatology/European League Against Rheumatism (ACR/EULAR) [9], after they signed a written consent and with ethical permission provided by the National Public Health and Medical Officer Service (21390-6-2017/EÜIG). Female patients with a high titer of anti-CCP antibodies were recruited for the study. Patients who had received biological therapy within three months of providing their blood sample were excluded from the study. Age- and sex-matched control sera were obtained from healthy volunteers at the University, who had not been vaccinated in the last three months and had no inflammatory or autoimmune diseases.

Blood was taken into Vacuette^®^ Z serum clot activator tubes (Greiner Bio-One). The tubes were left at room temperature for one hour, and then the samples were centrifuged at 800× *g* at 4 °C for 10 min. The serum was then aspirated, aliquoted, and stored at −20 °C until further use.

### 4.2. IgG Isolation and ACPA Purification

Total IgG isolation and subsequent ACPA purification were performed as previously described [46,66]. Briefly, IgG was isolated from serum samples using HiTrap Protein G affinity purification columns (Sigma, St. Louis, MO, USA) according to the manufacturer’s instructions, the protein concentration was adjusted to 1 mg/mL, and then the samples were dialyzed against PBS. The dialyzed IgG-containing samples were stored at −20 °C. ACPA was affinity purified from the IgG fraction of RA patients using citrulline-containing peptides (provided by Anna Magyar and Katalin Uray at ELTE Peptide Chemistry Research Group) immobilized on NHS (n-hydroxi-succinimide) columns (Hi Trap, GE Healthcare, Uppsala, Sweden). The flow-through fractions (FT, non-ACPA IgG) were collected then the citrulline peptide-specific ACPA were eluted at acidic pH. The FT and the eluate concentrations were determined and adjusted to the 0.5–1 mg/mL range. The samples were then dialyzed against 50 mM NH_4_HCO_3_ buffer for 2 h at room temperature and then with a fresh batch of buffer, overnight at 4 °C. The dialyzed fractions were tested for IgG concentration and citrulline-peptide specificity, then stored at −80 °C until further use.

### 4.3. Materials Used for Digestion and Chromatographic Separation

Unless otherwise stated, LC-MS grade solvents and reagents were purchased from Merck (Darmstadt, Germany). 1,4-dithiothreitol (DTT) reducing agent was obtained from Roche Diagnostics (Mannheim, Germany) and iodoacetamide (IAA) alkylating agent was purchased from Fluka Chemie (Buchs, Switzerland). RapiGest SF (lyophilized sodium-3-[(2-methyl-2-undecyl-1,3-dioxolan-4-yl)-methoxyl]-1-propane-sulfonate) denaturant was purchased from Waters (Milford, MA, USA). Mass spectrometry grade trypsin and Trypsin/Lys-C Mix were supplied by Promega (Madison, WI, USA).

### 4.4. Enzymatic Digestion

Samples collected from the ACPA and total serum IgG isolation procedures were digested in-solution as previously described [67]. Briefly, 33 pmol IgG isolate per sample was solubilized, then denatured by the addition of 2 μL 0.5% RapiGest SF and disulfide bonds were reduced by adding 5 μL 100 mM DTT for 30 min at 60 °C. Alkylation was accomplished by incubating for 30 min in the dark at room temperature with 2.5 μL 200 mM IAA and 5 μL 200 mM NH_4_HCO_3_. To improve peptide digestion over trypsin alone, Trypsin/Lys-C Mix was applied first in 1:100 ratio for 1 h at 37 °C, then trypsin was added in 1:25 ratio for 3 h at 37 °C. The reaction was stopped by adding formic acid to a final concentration of 6% (*v*/*v*). The samples were then dried, using vacuum centrifugation, and stored at −25 °C until the LC-MS measurement.

### 4.5. Nano LC-MS(MS)

Liquid chromatography separation was performed on an Ultimate 3000 nanoRSLC system (Dionex, Waltham, MA, USA). Samples were desalted using an Acclaim PepMap100 C-18 trap column (5 μm, 100 Å, 100 μm × 20 mm; Thermo Scientific, Sunnyvale, CA, USA) and peptides were separated by an Acquity UPLC M-Class Peptide BEH C18 column (1.7 μm, 130 Å, 75 μm × 250 mm; Waters, Milford, MA, USA). Gradient elution was applied with a flow rate of 300 nL/min at 48 °C (4% B from 0 to 11 min, followed by a 90 min gradient to 50% B, A: water + 0.1% formic acid, B: acetonitrile + 0.1% formic acid). For the mass spectrometry measurements, coupled to the nanoLC system, a Maxis II ETD Q-TOF (Bruker Daltonics, Bremen, Germany) equipped with a CaptiveSpray nanoBooster ion source was used in positive ionization mode. Capillary voltage was set to 1300 V, the nanoBooster pressure was 0.2 bar, the drying gas was heated to 150 °C, and the flow rate was 3 L/min.

The mass spectrometer was operated with the following ion-transfer tuning parameters: Funnel 1 RF 400 Vpp, Multipole RF 800 Vpp, quadrupole ion energy 4 eV, collision cell collision energy 7 eV, prepulse storage 10 μs. Glycopeptide identifications were conducted by recording the MS spectra over the mass range of m/z 150−3000 at 2 Hz. Triply charged precursor ions were selected to CID analysis performed at 4 Hz for abundant precursors (>25,000 cts/s) and at 0.5 Hz for low-abundance ones (>2500 cts/s). Glycopeptide quantification was based on MS experiments performed over the mass range of *m*/*z* 300−3000 at 1 Hz. Some samples were measured in replicates for quality control purposes.

The performance of the LC and MS instruments were validated using enolase and HeLa tryptic digest standards.

### 4.6. Data Evaluation

Raw data were recalibrated using Compass DataAnalysis 4.3 (Bruker Daltonics, Bremen, Germany). Components were identified based on their retention time, *m*/*z*, charge state and isotope pattern. Glycopeptides corresponding to the EEQYNSTYR peptide sequence were considered IgG1 glycoforms, those corresponding to EEQFNSTFR sequence were considered IgG2 glycoforms, while glycopeptides of the sequences EEQYNSTFR and EEQFNSTYR were considered IgG3 and IgG4, respectively. N-glycopeptides of all four subclasses of IgG were measured; however, due to their isobaric nature, IgG3 and IgG4 could not be distinguished with the measurement method used, thus they were handled together as IgG3/4. For samples that were measured in replicates, the average of the replicate results was used. We included those glycoforms in the study that were reliably (at least in two thirds of the samples) quantifiable. The exact list of investigated N-glycans, consisting of 19–19 glycoforms of IgG1, IgG2 and 8 glycoforms of IgG3/4, can be found in Appendix A. The average glycosylation pattern of the ACPA fraction of the age-matched patients is illustrated in Figure 1. Quantitative analysis of N-glycopeptides was conducted using our in-house developed software, GlycoPattern v4.7_b30, and was based on the area under the curve (AUC) of Gaussian peaks, fitted over the extracted ion chromatogram of the identified components in the MS1 chromatograms [68]. The following settings were used: min. number of isotopes to consider—5, peak split sigma—15, isotope cluster separation—1.5% and mass precision—10 ppm. The relative intensities were normalized to the sum of the total IgG glycopeptide abundances. The degree of galactosylation, sialylation, fucosylation, and the frequency of bisecting N-acetylglucosamine residues were calculated in the case of the total IgG, and also for each of the IgG isoforms separately, according to the following equation:(1)$$Glycosylation\;parameter=\frac{\sum_{i}c_{i}I_{i}}{\sum_{i}I_{i}}$$
where $I_{i}$ stands for the normalized intensity of the i-th glycoform, and $c_{i}$ is the contribution coefficient of the same glycoform. Information regarding the contributions of the abundance (coefficient) of each N-glycopeptide in the calculation of the four glycosylation features can be found in Appendix A.

Statistical analysis was performed using the R 4.3.0 environment with RStudio 2023.06.1 Build 524 [69,70]. Individual level comparisons between the ACPA and non-ACPA data were done using the paired, two-sided Welch’s *t*-test; while for groupwise comparisons between pairs of sample groups, the unpaired, two-sided Welch’s *t*-test was used, utilizing the *t*.test() function with the var. equal argument set to FALSE.

## 5. Conclusions

We investigated the glycosylation pattern of ACPA and non-ACPA IgG from RA patients, and compared the results with total IgG of healthy volunteers, in an isoform-specific manner.

We confirmed that ACPA IgG from RA patients displays a significantly lower level of galactosylation; however, this difference is completely absent for the IgG2 isoform. An increase in the frequency of the bisecting GlcNAc unit is also apparent for the non-ACPA IgG fraction compared to both the ACPA IgG and healthy control IgG samples, mainly driven by the IgG1 isoform. Furthermore, ACPA IgG1 displayed increased core-fucosylation compared to the IgG of healthy controls, while in the case of IgG2, the difference is in the opposite direction.

Differences on the group level between the ACPA and non-ACPA IgG fraction were reinforced by paired analysis of the same patients’ ACPA and non-ACPA samples. Galactosylation and sialylation of the IgG1 and IgG2 subclasses of ACPA, as well as non-ACPA IgG, has shown a significant, moderate negative correlation with the inflammatory markers CRP and ESR, as well as RF. However, no correlation was observed with DAS or CCP. Furthermore, a negative correlation between glycosylation and DAS was shown by the non-ACPA IgG fractions.

Our research expands the existing knowledge of the molecular foundations that contribute to the elucidation of the pathogenesis of rheumatoid arthritis. The results align well with the earlier findings and support both the application of age-matched patient groups and the importance of independently analyzing the different IgG subclasses. This research, however, is subject to some limitations. A larger number of patients in future studies would significantly improve the current results by minimizing the variance between donors, which would improve the power of the statistical analysis.

## Figures and Tables

**Figure 1 ijms-26-09626-f001:**
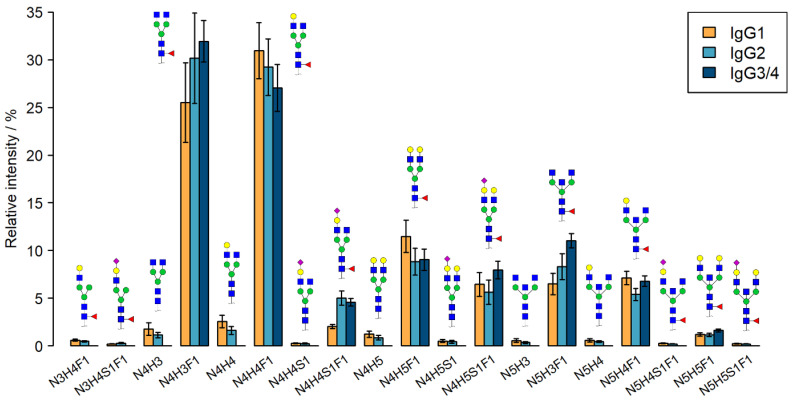
Average, isoform-specific N-glycosylation pattern of the ACPA fraction of RA patients. Intensities are normalized so that the sum of all relative intensities adds up to 100% for each IgG isoform separately. Error bars represent standard deviation. Note that for IgG1 and IgG2 all 19 glycopeptides were measured, but for IgG3/4 only 8 glycoforms were included. The coloured figures above the bars represent the structure of N-glycans (yellow circle: galactose, blue square: N-acetylglucosamine, green circle: mannose, red triangle: fucose, purple diamond: sialic acid) and the *X*-axis data show the composition of the sugars. N: N-acetylglucosamine, H: Hexose, S: Sialic acid, F: Fucose.

**Figure 2 ijms-26-09626-f002:**
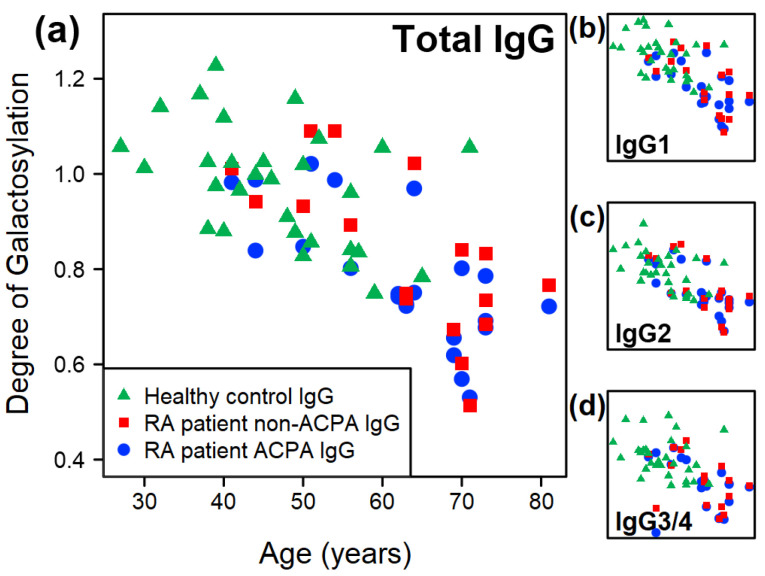
The relationship between the degree of galactosylation and age (**a**) across all IgG isoforms, and in the case of (**b**) IgG1, (**c**) IgG2, and (**d**) IgG3/4. All subplots are drawn using the same axis limits.

**Figure 3 ijms-26-09626-f003:**
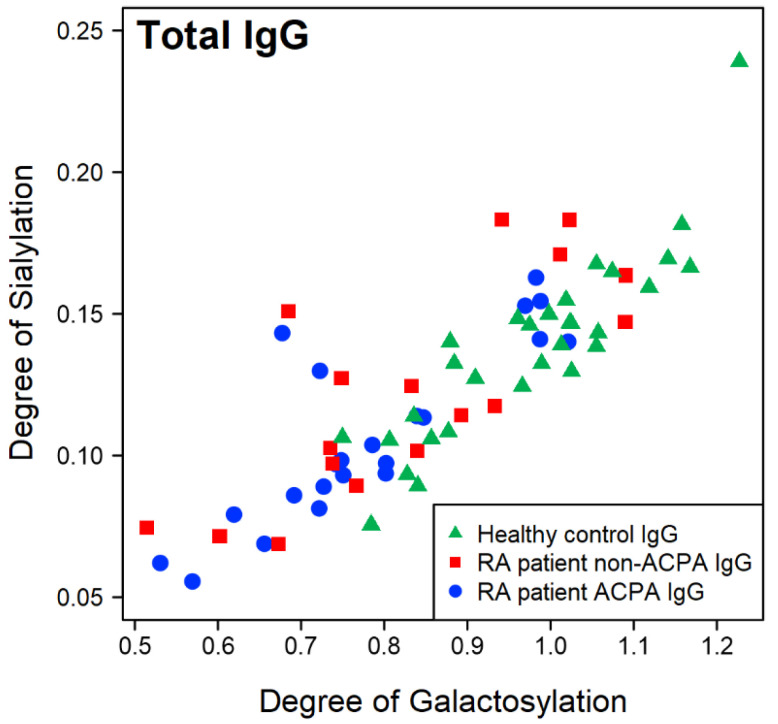
The relationship between the degree of galactosylation and the degree of sialylation across all IgG isoforms.

**Figure 4 ijms-26-09626-f004:**
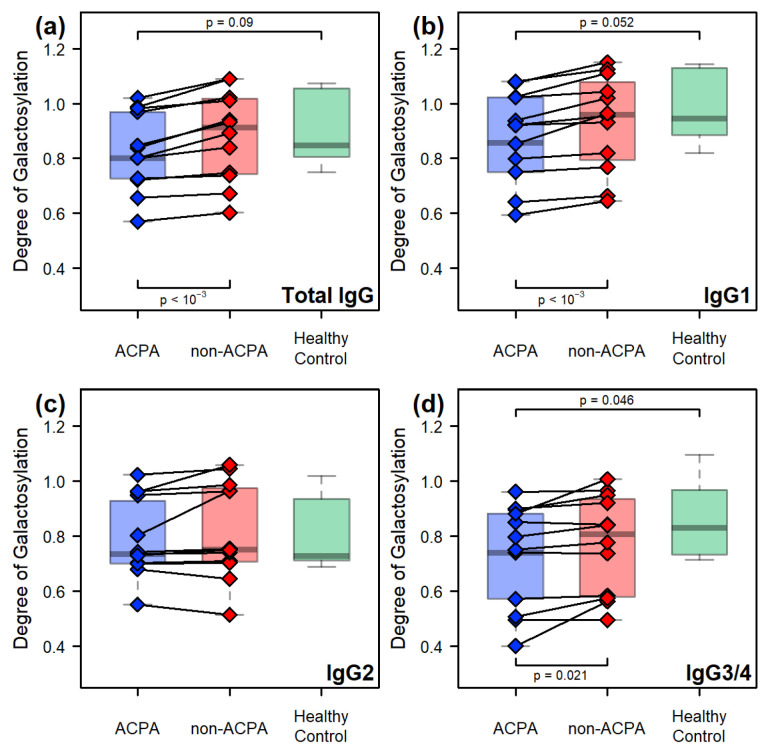
Comparisons of the degree of galactosylation between the sample groups: (**a**) across all IgG isoforms, and in the case of (**b**) IgG1, (**c**) IgG2, and (**d**) IgG3/4. The *p*-values found at the top of the subplots refer to comparisons between the sample groups, using Welch’s unpaired two-sample *t*-test, while those found at the bottom refer to the paired comparison made between the ACPA and non-ACPA fractions of the same RA patients, using Welch’s paired two-sample *t*-test.

**Figure 5 ijms-26-09626-f005:**
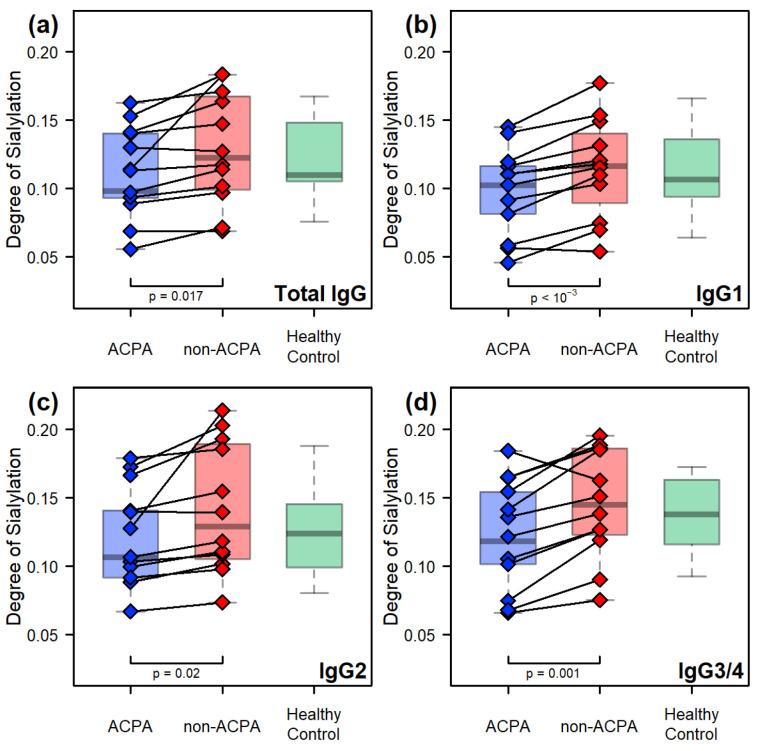
Comparisons of the degree of sialylation between the sample groups: (**a**) across all IgG isoforms, and in the case of (**b**) IgG1, (**c**) IgG2, and (**d**) IgG3/4. The *p*-values refer to the paired comparison made between the ACPA and non-ACPA fractions of the same RA patients, using Welch’s paired two-sample *t*-test.

**Figure 6 ijms-26-09626-f006:**
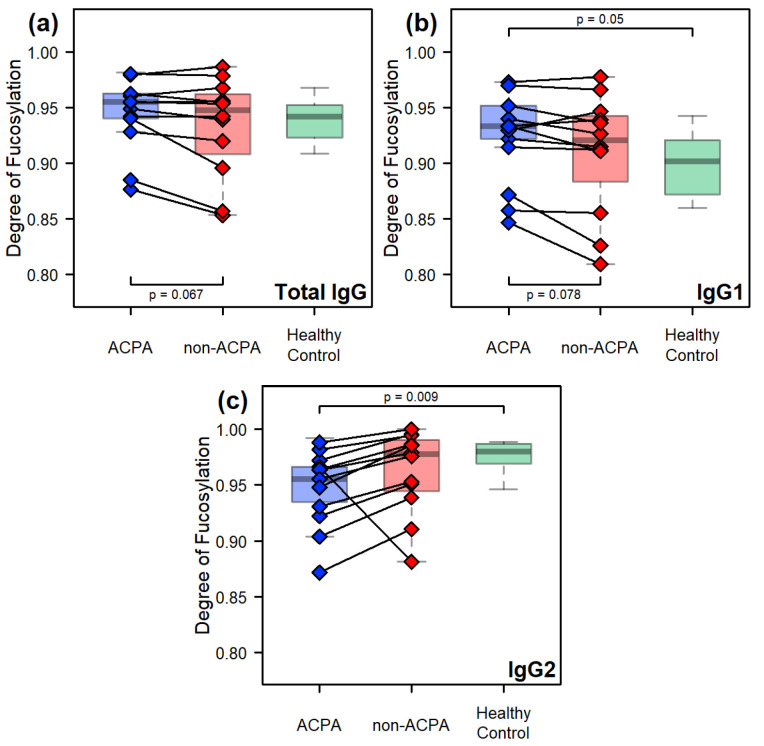
Comparisons of the degree of fucosylation between the sample groups: (**a**) across all IgG isoforms, and in the case of (**b**) IgG1, (**c**) IgG2. The *p*-values found at the top of the subplots refer to comparisons between the sample groups, using Welch’s unpaired two-sample *t*-test, while those found at the bottom refer to the paired comparison made between the ACPA and non-ACPA fractions of the same RA patients, using Welch’s paired two-sample *t*-test.

**Figure 7 ijms-26-09626-f007:**
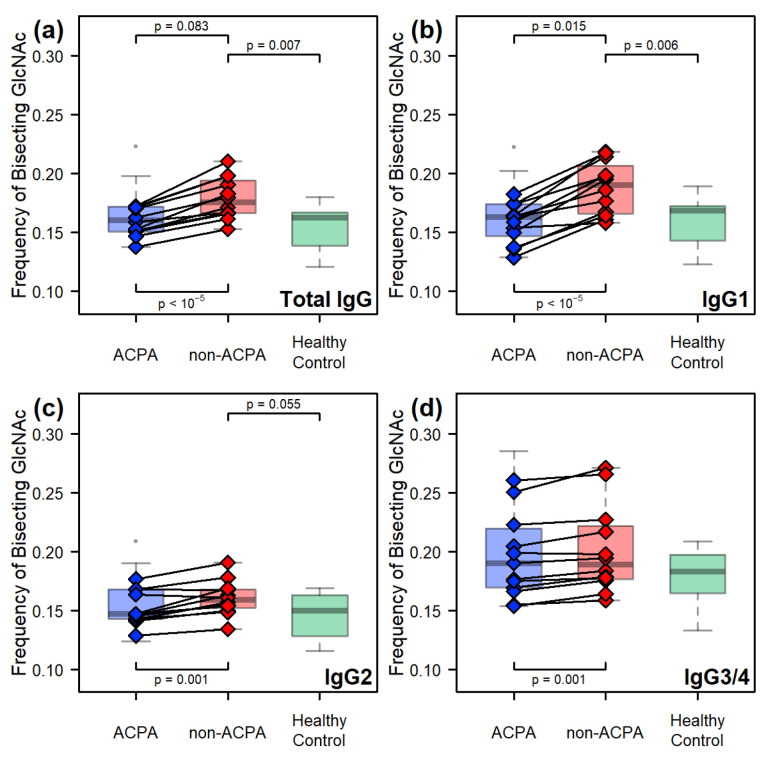
Comparisons of the frequency of the bisecting GlcNAc between the sample groups: (**a**) across all IgG isoforms, and in the case of (**b**) IgG1, (**c**) IgG2, and (**d**) IgG3/4. The *p*-values found at the top of the subplots refer to comparisons between the sample groups, using Welch’s unpaired two-sample *t*-test, while those found at the bottom refer to the paired comparison made between the ACPA and non-ACPA fractions of the same RA patients, using Welch’s paired two-sample *t*-test.

**Figure 8 ijms-26-09626-f008:**
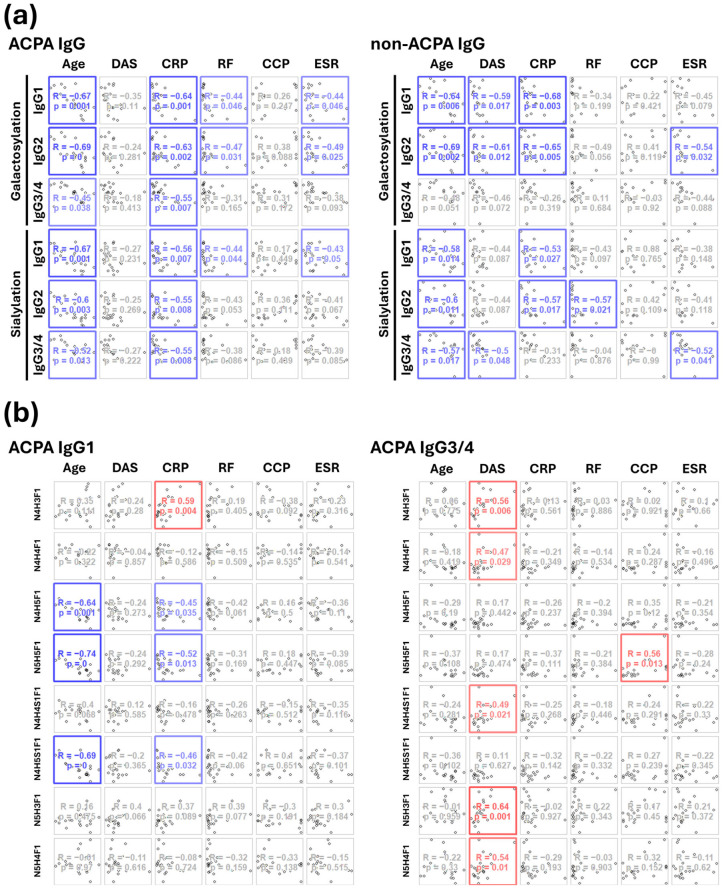
Correlation plots of age and clinical parameters with (**a**) the degree of isoform specific galactosylation and sialylation of ACPA and non-ACPA IgG, and (**b**) the relative abundance of eight glycoforms, which were all measured both in the case of IgG1 and IgG3/4 isoforms. Negative correlations are represented in blue, positive correlations are represented in red.

## Data Availability

All generated LC-MS data used in this study is available at the MassIVE repository (ID: MSV000096838). https://massive.ucsd.edu/ProteoSAFe/static/massive.jsp, accessed on 17 September 2025. Username for web access: MSV000096838_reviewer. Password: RAglyco2025.

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
