# Peer review of "Immunoglobulin G Subclass-Specific Glycosylation Changes in Rheumatoid Arthritis"

_ijms, 2025, doi:10.3390/ijms26199626_

Round 1

Reviewer 1 Report

Comments and Suggestions for Authors

This is an interesting paper that investigates subclass-specific IgG glycosylation differences in rheumatoid arthritis, comparing ACPA, non-ACPA, and healthy controls. The research provides insights into how galactosylation, sialylation, and fucosylation vary across IgG subclasses and correlate with clinical markers. Below are some contributions to make the paper more fit for publication. 

The authors should explain why only female RA patients were included, and also, make this information clear in the abstract. 

Why controls were excluded if vaccinated within three months? Vaccination may influence immune activation, but why this cutoff was chosen is not clear.

The age-matched groups have small numbers of participants. It makes the data hardly generalizable. I believe that eventual limitations should be explained here, but also, the authors should consider adjusting the paper for a short communication

Did the authors validated the LC and Q-ToF conditions? Was the separation efficiency quantified? This information should be considered.

Did the authors considered any other method for properly distinguishing IgG3 and IgG4 such as additional enzymatic digestion?

In statistics section, the authors only stated that R was used, but does not specify which tests were applied. Please provide a full description of the chosen tests.

Terms such as  “noticeable.” are not adequate. Please use words that help to  distinguish between trends and significant findings; provide confidence intervals wherever possible.

Many correlations with age or disease markers have low R² (e.g., R² ≤ 0.19 for fucosylation/bisecting GlcNAc. Such correlations should not be presented as biologically relevant. 

in figure 1, please consider making it fully understandable as is, for instance, figure two. Provide information on all the colors beyond the bars and the X axis data.

Reviewer 2 Report

Comments and Suggestions for Authors

Abbreviations in the Abstract are not properly explained – for example DAS.

The abstract is difficult-to-read. The authors should revise it to become more reader-friendly. For that purpose, the abstract needs to be structured according to the journal guidelines for original articles – aim, patients and methods, results…

Keywords are not appropriate. Please, use the MeSH terms for choosing the keywords.

The introductory part needs to be revised. The authors should emphasize more on the clinical implications of the presented theoretical data, not just present random facts.

The Results section needs to be revised, because it contains comparison with the results from other similar studies, which need to be moved into the Discussion section.

Inclusion and exclusion criteria are not properly listed.

The whole manuscript is not well-structured. Materials and methods section is after the Results and Discussion section.

The authors should highlight the novelty of their study.

Conclusion section is too long. The authors should summarize the results obtained in the current study, not just repeat everything from the previous sections in the Conclusion.

Round 2

Reviewer 2 Report

Comments and Suggestions for Authors

The authors have revised the manuscript in accordance with the reviewers' suggestions.